# Is maternal weight gain between pregnancies associated with risk of large-for-gestational age birth? Analysis of a UK population-based cohort

Nida Ziauddeen,[1] Sam Wilding,[1] Paul J Roderick,[1] Nicholas S Macklon,[2,3] Nisreen A Alwan[1,4]

¹School of Primary Care and Population Sciences, Faculty of Medicine, University of Southampton, Southampton, UK
²Department of Obstetrics and Gynaecology, University of Copenhagen, Zealand University Hospital, Roskilde, Denmark
³London Women's Clinic, London, UK
⁴NIHR Southampton Biomedical Research Centre, University of Southampton and University Hospital Southampton NHS Foundation Trust, Southampton, UK

**Correspondence to**
Nida Ziauddeen;
N.Ziauddeen@soton.ac.uk

## ABSTRACT

**Objective** Maternal overweight and obesity during pregnancy increases the risk of large-for-gestational age (LGA) birth and childhood obesity. We aimed to investigate the association between maternal weight change between subsequent pregnancies and risk of having a LGA birth.

**Design** Population-based cohort.

**Setting** Routinely collected antenatal healthcare data between January 2003 and September 2017 at University Hospital Southampton, England.

**Participants** Health records of women with their first two consecutive singleton live-birth pregnancies were analysed (n=15 940).

**Primary outcome measure** Risk of LGA, recurrent LGA and new LGA births in the second pregnancy.

**Results** Of the 15 940 women, 16.0% lost and 47.7% gained weight (≥1 kg/m²) between pregnancies. A lower proportion of babies born to women who lost ≥1 kg/m² (12.4%) and remained weight stable between −1 and 1 kg/m² (11.9%) between pregnancies were LGA compared with 13.5% and 15.9% in women who gained 1–3 and ≥3 kg/m², respectively. The highest proportion was in obese women who gained ≥3 kg/m² (21.2%). Overweight women had a reduced risk of recurrent LGA in the second pregnancy if they lost ≥1 kg/m² (adjusted relative risk (aRR) 0.69, 95% CI 0.48 to 0.97) whereas overweight women who gained ≥3 kg/m² were at increased risk of new LGA after having a non-LGA birth in their first pregnancy (aRR 1.35, 95% CI 1.05 to 1.75). Normal-weight women who gained weight were also at increased risk of new LGA in the second pregnancy (aRR 1.26, 95% CI 1.06 to 1.50 with gain of 1–3 kg/m² and aRR 1.34, 95% CI 1.09 to 1.65 with gain of ≥3 kg/m²).

**Conclusions** Losing weight after an LGA birth was associated with a reduced LGA risk in the next pregnancy in overweight women, while interpregnancy weight gain was associated with an increased new LGA risk. Preventing weight gain between pregnancies is an important measure to achieve better maternal and offspring outcomes.

## INTRODUCTION

The prevalence of maternal obesity has been rising over time. It has more than doubled in England between 1989 and 2007 (7.6%–15.6%), with the proportion of normal weight pregnancies showing a 12% decrease from 65.6% to 53.6%.[1] Maternal overweight and obesity is a key risk factor for adverse maternal and birth outcomes. It also increases the risk of long-term health problems in the child including obesity, cardiovascular disease, diabetes and cognitive and behavioural disorders.[2] Birth weight is a key early life predictor of long-term health outcomes such as obesity and cardiovascular disease[3] and potentially acts as a mediator on the causal pathway between maternal obesity and long-term offspring outcomes. The incidence of large-for-gestational age (LGA) birth, defined as >90th percentile weight for gestational age, has increased over time in high-income countries.[4 5] LGA is associated with both childhood[6 7] and adult obesity.[8–10] A key risk factor for LGA birth is gestational diabetes (GDM),[11] the incidence of which has also increased over time.[12 13] Offspring of mothers with GDM have increased risk of childhood overweight and obesity.[14 15] Maternal obesity is an established risk factor for both GDM and LGA birth.[16] Change in maternal body mass index (BMI) between pregnancies could modify the risk of LGA birth in the subsequent pregnancy.

**Strengths and limitations of this study**

► Utilises antenatal care and birth data from a large population-based cohort including women from all socioeconomic backgrounds.
► Objective measurement of both exposure (maternal weight) and outcome in two pregnancies per woman.
► Self-reported data for covariates.
► Lack of information on breastfeeding duration and maternal weight gain during pregnancy.

Birth weight, on average, increases with parity. First-born infants tend to have the lowest birth weight among their younger siblings[17–19] up to the fourth pregnancy.[20] However, birth weight was found to decrease with parity for women who had short intervals between their pregnancies (<12 months) while the increase in birth weight with parity was more pronounced in women with long intervals (>24 months).[20] Also, maternal weight change between pregnancies was found to modify the relationship between parity and birth weight. Women who returned to their prepregnancy weight before the next conception had infants who weighed less than infants of women who retained or gained weight between pregnancies.[20] In a UK-based study, women who lost at least 6 kg between their first and second pregnancy had a smaller average increase in birth weight of the second baby compared with women who gained 10 kg or more (in a 1.60 m tall woman, 6 kg equates to ~2.3 kg/m² and 10 kg to ~3.8 kg/m²).[18]

A large US study showed that women were at an increased risk of having an LGA baby in the second pregnancy if their prepregnancy BMI category increased towards overweight or obese between their first and second pregnancies. This applied to all first pregnancy BMI categories, except underweight women who became normal weight by the start of their second pregnancy. Overweight and obese women who dropped BMI category by their second pregnancy remained at an increased risk of LGA birth, but had a lower risk compared with women whose BMI category increased between pregnancies.[21]

Another US-based study showed that interpregnancy weight gain of ≥2 kg/m² in obese women was associated with increased risk of LGA. Weight loss of ≥2 kg/m² was associated with a lower adjusted LGA risk compared with the women who maintained their weight within 2 kg/m² change between pregnancies.[22]

Two studies found a reduced risk of 'new' LGA in the second pregnancy following a non-LGA birth in the first pregnancy with interpregnancy weight loss of >1 kg/m², and an increased risk with modest (1–3 kg/m²) and large (≥3 kg/m²) weight gain. In stratified analysis, the association was stronger in women with a first pregnancy BMI of <25 kg/m².[23 24] A third study only found an increased risk of new LGA in normal weight women who gained ≥4 kg/m² between pregnancies and no association in overweight women.[25]

To our knowledge, only one study has examined the risk of recurrent LGA (occurring in both first and second pregnancies) in relation to maternal weight change between pregnancies.[26] The study, conducted in Aberdeen, Scotland, included 24 520 women of which 813 women had LGA births in both pregnancies. Interpregnancy weight gain (≥2 kg/m²) was associated with increased risk of recurrent LGA, while weight loss (≥2 kg/m²) was protective. Women with BMI <25 kg/m² were at increased risk of recurrent LGA on gaining weight whereas women with BMI ≥25 kg/m² were at reduced risk of recurrent LGA on losing weight.[26]

In this study, we aimed to investigate the association between the incidence of LGA, recurrent LGA and new LGA births in the second pregnancy and maternal change in BMI between the first and second pregnancies, stratifying by maternal BMI category in the first pregnancy, in a population-based cohort in the South of England.

## METHODS

This is a population-based cohort of prospectively collected routine healthcare data for antenatal care between January 2003 and September 2017 at University Hospital Southampton, Hampshire, UK. This included all women registered for maternity care at this hospital (n=82 098 pregnancies), which is a regional centre for maternity care in and around Southampton. Records of women with their first two consecutive singleton live birth pregnancies were included. Records with unfeasible weight (<30 kg), height (>2 m) and gestational age (>301 days) values were excluded.

### Exposure assessment

Maternal weight in kilograms was routinely measured by a midwife at the first antenatal (booking) appointment of each pregnancy, which is recommended to take place ideally by 10 weeks gestation in the UK, according to the National Institute for Health and Care Excellence Guidelines.[27] Any woman who had a booking appointment at or after 24 weeks of pregnancy was excluded. Height was self-reported. BMI was calculated as weight (in kg) divided by height (in metres) squared.

BMI at the start of the first pregnancy was categorised as underweight (BMI <18.5 kg/m²), normal weight (18.5–24.9 kg/m²), overweight (25.0–29.9 kg/m²) and obese (≥30 kg/m²). Change in BMI was calculated as the difference in BMI measured at the booking appointments of the first two consecutive live birth pregnancies for each woman. This change in BMI was then categorised as weight loss (≥1 kg/m²), weight stable (−1 to 1 kg/m²) and two categories of weight gain (1–3 and ≥3 kg/m²).

### Outcome assessment

Birth weight (grams) was measured by healthcare professionals at birth as part of routine care. Gestational age was based on a dating ultrasound scan which routinely takes place between 10 and 13 weeks of gestation.[27] Age- and sex-specific birth weight centiles were calculated using reference values for England and Wales provided in the most recently released national data.[28] LGA was defined as >90th percentile weight for gestational age. This was only defined for babies born between 24 and 42 weeks of gestation as reference values only exist for these gestational ages and with determinate sex.

### Covariates

Maternal date of birth is recorded at the booking appointment and converted to age (in years) on extraction of the dataset to maintain anonymity. Highest maternal

educational qualification was self-reported and categorised as primary, secondary, college, undergraduate, postgraduate, graduate and none. For the purposes of this analysis, this was condensed to three categories— secondary (General Certificate of Secondary Education, GCSE) and under, college (A levels) and university degree or above. Self-reported ethnicity was recorded under 16 categories and condensed to White, Mixed, Asian, Black/African/Caribbean and Other. Categories of not asked and not stated were coded as missing. Smoking was self-reported as current smoking or non-smoking. Non-smokers were further asked if they had ever smoked or had previously smoked and quit. This was categorised as stopped >12 months before conception, stopped <12 months before conception or stopped when pregnancy confirmed. Employment status was self-reported at booking appointment and categorised as employed, unemployed, in education and not specified. Infertility treatment was categorised as no/investigations only and yes (hormonal only, in vitro fertilisation, gamete intrafallopian transfer and other surgical) in either one or both pregnancies. In this population, an oral glucose tolerance test was used for screening for GDM in women with one or more risk factors (BMI >30 kg/m$^2$; GDM in previous pregnancy; previous baby weighing ≥4.5 kg; diabetes in parents or siblings and of Asian, African-Caribbean or Middle Eastern ethnicity).[29] GDM diagnosis was then reported in the database. Interpregnancy interval was defined as the interval between the first live birth and conception of the second pregnancy. The difference in days between two consecutive live births was calculated and gestational age of the latter birth subtracted from this to derive the interpregnancy interval.

## Statistical analysis

All analysis was performed using Stata V.15.[30] Univariable comparisons were carried out using Analysis of variance (ANOVA) for continuous variables and $\chi^2$ test for categorical variables. Generalised linear regression with log link[31] was used to examine the association between the categorised variable of maternal change in BMI between pregnancies with risk of LGA in the second pregnancy. This was analysed first in the whole sample and then stratified by 'baseline' maternal BMI category as calculated in the first antenatal appointment of the first pregnancy.

Risk of LGA in the second pregnancy was explored in the whole sample adjusting for previous pregnancy outcome of LGA. The risk of new LGA in second pregnancy after having a non-LGA baby in the first pregnancy was explored in the subsample of women who had non-LGA births in the first pregnancy. The risk of recurrent LGA (LGA in both pregnancies) was explored in a subsample of women who had LGA births in the first pregnancy.

Initial univariable analysis was followed by multivariable models adjusting for potential confounding factors— maternal age, ethnicity, highest educational qualification, whether or not undergone infertility treatment, employment status, smoking behaviour in second pregnancy, baseline BMI, GDM in second pregnancy and interpregnancy interval. Sensitivity analysis was conducted adding gestational age at booking in the second pregnancy to the models.

A statistical significance level of 0.05 with 95% CI was used in the regression models.

## Ethical considerations

All data were fully anonymised by the data holder before being accessed by the research team.

## Patient and public involvement

Patients and public were not involved in setting the research question or the outcome measures, nor were they involved in developing plans for the design or implementation of the study. However, pregnant woman and mothers of young children have been involved in the planning stages of a research project building on this analysis.

## RESULTS

The first and second pregnancies of 15 940 women were included. Of these, 16.0% of women lost ≥1 kg/m$^2$, 36.3% remained weight stable (−1 to 1 kg/m$^2$), 27.9% gained 1–3 kg/m$^2$ and 19.8% gained ≥3 kg/m$^2$ between their first and second live birth pregnancies. Weight loss of >2 kg/m$^2$ was observed in 7.3% of women whereas 30.5% gained >2 kg/m$^2$. Mean BMI at second pregnancy booking was 30.8 kg/m$^2$ (SD 5.9) in women who gained ≥3 kg/m$^2$, 25.9 kg/m$^2$ (SD 4.7) in women who gained 1–3 kg/m$^2$, 24.1 kg/m$^2$ (SD 5.1) in women who lost weight and 23.8 kg/m$^2$ (SD 4.4) women whose weight remained stable between pregnancies (p<0.001) (table 1).

Women who gained ≥3 kg/m$^2$ by the start of their second pregnancy were more likely to be smokers, unemployed, with lower educational attainment and to have a longer interpregnancy interval, compared with those who maintained a stable weight between pregnancies. Mean maternal age was lowest in the women who gained ≥3 kg/m$^2$ (27.3 years, SD 5.5) and highest in the women who remained weight stable (29.8 years, SD 5.3). Mean maternal age in women who lost weight was 28.7 years (SD 5.4).

Mothers who gained ≥3 kg/m$^2$ were more likely to be obese (48.3%) at the start of the second pregnancy compared with 16.1% in women who gained 1–3 kg/m$^2$, 9.2% in women who remained weight stable and 11.9% in women who lost ≤1 kg/m$^2$.

Figure 1 shows the percentage of women in each BMI category in the first and second pregnancy and the weight gain over time. There has been a decline in normal weight women at first pregnancy and a slight increase in overweight and obese women over time. There also was a slight decline in the percentage of women gaining ≥3 kg/m$^2$ and a slight increase in those gaining 1–3 kg/m$^2$.

**Table 1** Maternal and birth characteristics in the second live birth pregnancy categorised by maternal weight change gain from the first live birth pregnancy for the period of January 2003 to September 2017, University Hospital Southampton NHS Foundation Trust, Hampshire, England

| | Lost ≤ −1 kg/m² from previous pregnancy | Weight stable (> −1 to <1 kg/m²) | Gained 1–3 kg/m² from previous pregnancy | Gained ≥3 kg/m² from previous pregnancy | P value* |
|---|---|---|---|---|---|
| N | 2548 | 5785 | 4446 | 3161 | |
| Maternal age, years (mean±SD) | 28.7±5.4 | 29.8±5.3 | 29.2±5.4 | 27.3±5.5 | <0.001 |
| Timing of first booking appointment, weeks (mean±SD) | 10.8±2.3 | 11.0±2.3 | 11.1±2.4 | 11.0±2.6 | <0.001 |
| Maternal BMI at booking, kg/m² (mean±SD) | 24.1±5.1 | 23.8±4.4 | 25.9±4.7 | 30.8±5.9 | <0.001 |
| Maternal BMI at booking in first pregnancy (%, 95% CI) | | | | | |
| Underweight (<18.5) | 0.8 (0.5 to 1.2) | 4.3 (3.8 to 4.8) | 5.3 (4.7 to 6.0) | 3.7 (3.1 to 4.4) | <0.001 |
| Normal weight (18.5 to 24.9) | 47.6 (45.6 to 49.5) | 67.4 (66.2 to 68.6) | 62.5 (61.0 to 63.9) | 49.0 (47.2 to 50.7) | |
| Overweight (25.0 to 29.9) | 30.1 (28.3 to 31.9) | 19.4 (18.4 to 20.5) | 22.0 (20.8 to 23.3) | 29.5 (28.0 to 31.2) | |
| Obese (≥30.0) | 21.5 (19.9 to 23.2) | 8.9 (8.2 to 9.7) | 10.2 (9.3 to 11.1) | 17.8 (16.5 to 19.2) | |
| Maternal BMI at booking in second pregnancy (%, 95% CI) | | | | | |
| Underweight (<18.5) | 6.9 (5.9 to 7.9) | 4.3 (3.8 to 4.8) | 0.6 (0.4 to 0.9) | 0.0 (0.0 to 0.2) | <0.001 |
| Normal weight (18.5 to 24.9) | 61.1 (59.2 to 63.0) | 66.8 (65.6 to 68.1) | 50.7 (49.2 to 52.1) | 14.9 (13.7 to 16.2) | |
| Overweight (25.0 to 29.9) | 20.1 (18.6 to 21.7) | 19.7 (18.7 to 20.7) | 32.6 (31.2 to 34.0) | 36.7 (35.0 to 38.4) | |
| Obese (≥30.0) | 11.9 (10.7 to 13.3) | 9.2 (8.5 to 10.0) | 16.1 (15.0 to 17.2) | 48.3 (46.6 to 50.1) | |
| Maternal smoking status at booking (%, 95% CI) | | | | | |
| Never smoked/quit | 57.2 (55.3 to 59.2) | 63.0 (61.8 to 64.3) | 60.5 (59.0 to 62.0) | 50.7 (48.9 to 52.4) | <0.001 |
| Stopped >1 year before conceiving | 16.1 (14.6 to 17.5) | 17.2 (16.3 to 18.2) | 17.7 (16.5 to 18.8) | 14.9 (13.7 to 16.2) | |
| Stopped <1 year prior to conceiving | 4.0 (3.3 to 4.8) | 2.8 (2.4 to 3.2) | 3.5 (3.0 to 4.1) | 4.9 (4.2 to 5.7) | |
| Stopped when pregnancy confirmed | 6.8 (5.8 to 7.8) | 5.9 (5.3 to 6.6) | 6.9 (6.2 to 7.7) | 10.3 (9.3 to 11.4) | |
| Continued smoking | 15.9 (14.5 to 17.4) | 11.0 (10.2 to 11.8) | 11.4 (10.5 to 12.4) | 19.1 (17.8 to 20.6) | |
| Maternal education (%, 95% CI) | | | | | |
| Secondary (GCSE) or under | 30.7 (28.9 to 32.5) | 24.0 (22.9 to 25.2) | 29.4 (28.1 to 30.8) | 36.3 (34.6 to 38.0) | <0.001 |
| College (A levels) | 40.4 (38.5 to 42.3) | 38.8 (37.6 to 40.1) | 39.5 (38.1 to 41.0) | 45.8 (44.0 to 47.5) | |
| University degree or above | 28.9 (27.2 to 30.7) | 37.1 (35.9 to 38.4) | 31.1 (29.7 to 32.5) | 17.9 (16.6 to 19.3) | |
| Maternal employment (%, 95% CI) | | | | | |
| Employed | 66.2 (64.3 to 68.0) | 71.7 (70.5 to 72.9) | 67.2 (65.8 to 68.5) | 56.5 (54.8 to 58.2) | <0.001 |
| Unemployed | 31.8 (30.0 to 33.7) | 26.9 (25.8 to 28.1) | 31.1 (29.7 to 32.5) | 41.6 (39.8 to 43.3) | |
| In education | 0.9 (0.6 to 1.4) | 0.8 (0.6 to 1.1) | 1.1 (0.8 to 1.4) | 1.3 (0.9 to 1.8) | |
| Not specified | 1.0 (0.7 to 1.5) | 0.6 (0.4 to 0.8) | 0.7 (0.5 to 1.0) | 0.6 (0.4 to 1.0) | |
| Ethnicity (%, 95% CI) | | | | | |
| White | 89.9 (88.7 to 91.1) | 88.0 (87.1 to 88.8) | 85.1 (84.0 to 86.1) | 84.8 (83.5 to 86.1) | <0.001 |
| Mixed | 0.8 (0.5 to 1.3) | 0.9 (0.7 to 1.2) | 1.4 (1.1 to 1.8) | 1.6 (1.1 to 2.0) | |
| Asian | 4.8 (4.0 to 5.7) | 5.6 (5.0 to 6.0) | 7.2 (6.5 to 8.0) | 7.7 (6.8 to 8.7) | |
| Black/African/Caribbean | 0.6 (0.4 to 1.0) | 1.0 (0.8 to 1.3) | 1.6 (1.3 to 2.1) | 2.4 (1.9 to 3.0) | |

Continued

**Table 1** Continued

| | Lost ≤ −1 kg/m² from previous pregnancy | Weight stable (> −1 to <1 kg/m²) | Gained 1–3 kg/m² from previous pregnancy | Gained ≥3 kg/m² from previous pregnancy | P value* |
|---|---|---|---|---|---|
| Other | 0.7 (0.4 to 1.1) | 1.0 (0.8 to 1.3) | 1.0 (0.8 to 1.4) | 1.3 (0.9 to 1.7) | |
| Not specified | 3.1 (2.5 to 3.9) | 3.5 (3.0 to 4.0) | 3.6 (3.1 to 4.2) | 2.2 (1.8 to 2.8) | |
| Interpregnancy interval (median, IQR) | 21.7 (14.4 to 32.7) | 21.6 (14.1 to 32.0) | 23.7 (14.4 to 35.6) | 27.7 (16.0 to 45.6) | <0.001 |
| Interpregnancy interval (%, 95% CI) | | | | | |
| 0–11 months | 17.4 (15.9 to 18.9) | 17.6 (16.6 to 18.6) | 18.1 (17.0 to 19.3) | 16.6 (15.4 to 17.9) | <0.001 |
| 12–23 months | 39.8 (37.8 to 41.7) | 39.9 (38.6 to 41.1) | 33.1 (31.7 to 34.5) | 26.3 (24.8 to 27.9) | |
| 24–35 months | 22.6 (21.0 to 24.2) | 23.6 (22.5 to 24.7) | 24.4 (23.2 to 25.7) | 20.5 (19.1 to 21.9) | |
| 36 months or more | 20.3 (18.7 to 21.9) | 18.9 (17.9 to 20.0) | 24.3 (23.1 to 25.6) | 36.5 (34.9 to 38.2) | |
| Birth weight, g (mean±SD) | 3463±563 | 3467±523 | 3507±536 | 3531±558 | |
| Previous size at birth (first pregnancy) | | | | | |
| Small-for-gestational age | 13.1 (11.8 to 14.4) | 12.6 (11.8 to 13.5) | 11.7 (10.8 to 12.7) | 12.4 (11.3 to 13.6) | 0.11 |
| Appropriate-for-gestational age | 79.6 (77.9 to 81.1) | 81.1 (80.0 to 82.1) | 81.2 (80.1 to 82.4) | 79.9 (78.4 to 81.3) | |
| Large-for-gestational age | 7.4 (6.4 to 8.5) | 6.3 (5.7 to 7.0) | 7.1 (6.3 to 7.8) | 7.7 (6.8 to 8.7) | |
| Size at birth (second pregnancy) | | | | | |
| Small-for-gestational age | 8.7 (7.6 to 9.8) | 7.0 (6.4 to 7.7) | 6.2 (5.5 to 6.9) | 6.7 (5.9 to 7.6) | <0.001 |
| Appropriate-for-gestational age | 79.0 (77.3 to 80.5) | 81.1 (80.0 to 82.1) | 80.3 (79.1 to 81.5) | 77.4 (75.9 to 78.9) | |
| Large-for-gestational age | 12.4 (11.1 to 13.7) | 11.9 (11.1 to 12.8) | 13.5 (12.5 to 14.5) | 15.9 (14.6 to 17.2) | |

*P values calculated using ANOVA for continuous and $\chi^2$ test for categorical variables.

The proportion of LGA births were higher in all BMI categories in the second pregnancy (figure 2). A lower proportion of babies born to women who lost weight (12.4%) or remained weight stable (11.9%) between pregnancies were LGA compared with 13.5% in women who gained 1–3 kg/m² and 15.9% in women who gained ≥3 kg/m² (p<0.001) (table 1, figure 3). Compared with normal weight women, overweight and obese women were at increased risk of LGA births in both pregnancies with risk highest in obese women (unadjusted relative risk (RR) 2.06, 95% CI 1.78 to 2.38 and 1.86, 95% CI 1.69 to 2.05 in first and second pregnancy, respectively). The lowest proportion of LGA births in the second pregnancy was in underweight women in the first pregnancy who remained weight stable (2.8%), while the highest was in obese women who gained ≥3 kg/m² (21.2%). Within BMI categories, recurrent LGA was lowest in normal weight and overweight women who lost weight and highest in obese women who gained 1–3 kg/m².

Women who gained ≥3 kg/m² were at increased risk of LGA in the second pregnancy in the full sample compared with remaining weight stable (adjusted relative risk (aRR) 1.28, 95% CI 1.14 to 1.44) (figure 3). There was a significantly reduced risk of recurrent LGA birth in the second pregnancy in overweight women who had a LGA infant in the first pregnancy and lost ≥1 kg/m² in weight (aRR 0.69, 95% CI 0.48 to 0.97) (table 2, online

supplementary figure 1). No association was observed between risk of recurrent LGA and maternal BMI change between pregnancies in underweight, normal weight and obese women.

There was an increased risk of new LGA birth in the second pregnancy after having a non-LGA infant in the first pregnancy in normal weight women who gained 1–3 kg/m² (aRR 1.26, 95% CI 1.06 to 1.50) and in normal weight and overweight women who had gained ≥3 kg/m² weight (aRR 1.34, 95% CI 1.09 to 1.65, aRR 1.35, 95% CI 1.05 to 1.75, respectively) (table 3, online supplementary figure 2). No association was observed between the risk of new LGA in the second pregnancy and maternal BMI interpregnancy change in obese women.

## DISCUSSION

This study examined the association between change in women's BMI between their first and second live birth pregnancies and risk of LGA birth in the second pregnancy in a population-based cohort of 15 940 women in the South of England. Almost half of the sample (48%) of women gained ≥1 kg/m² in the time between the first antenatal care visits during their first and second pregnancies. The proportion of LGA births was significantly higher in women with an interpregnancy weight gain of ≥3 kg/m² (16%) compared with women who lost weight (12%)

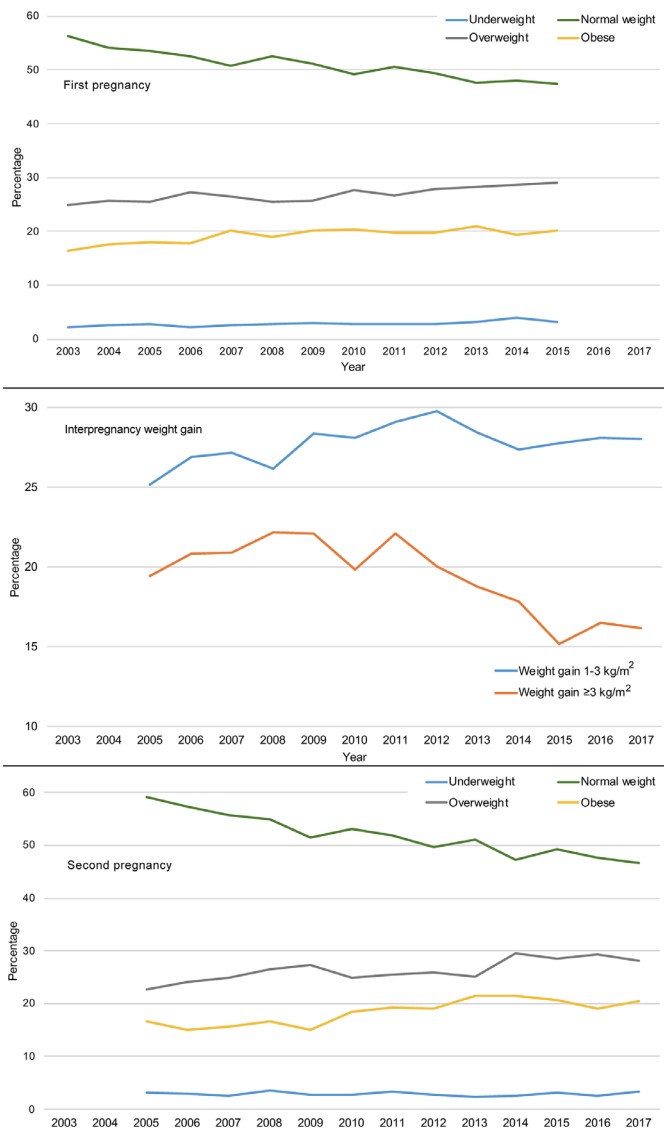

**Figure 1** The percentage of women in each body mass index (BMI) category in the first and second pregnancy and weight gain over time in the cohort (2003–2017).

and those who remained weight stable (12%) between pregnancies. Overweight women who lost ≥1 kg/m² had a reduced risk of recurrent LGA. Normal weight women who gained 1–3 kg/m² and both normal weight and overweight women who gained ≥3 kg/m² between pregnancies had an increased risk of LGA birth in their second pregnancy after a non-LGA birth in the first.

Compared with the population-based Swedish cohort which carried out a similar analysis for LGA and other outcomes in 151 025 women using data from 1992 to 2001, a lower proportion of women remained weight stable in our cohort (46% compared with 36%) and a higher proportion lost (11% compared with 16%) or gained (43% compared with 48%) weight. Among women who gained weight, a higher proportion gained ≥3 kg/m² in this cohort (20%) compared with the Swedish cohort (11%).[23] Similarly, in comparison to a population-based cohort of 24 520 women in Aberdeen, Scotland; for the

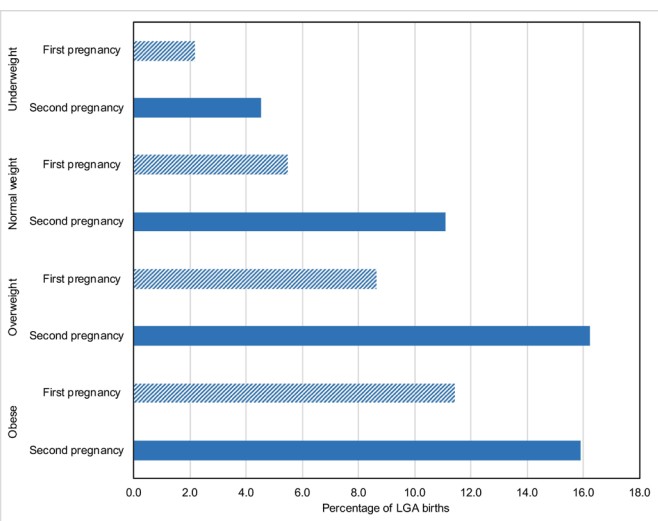

**Figure 2** The percentage of large-for-gestational age (LGA) births in first and second pregnancy by maternal body mass index category.

period 1986–2013, a larger proportion of women in our study both lost and gained weight.[26] The differences could reflect the increase in the prevalence of maternal overweight and obesity over time since our data are more recent.

In the adjusted model utilising the full sample, we showed an increased risk of LGA in the second pregnancy for interpregnancy weight gain compared with remaining weight stable. In a population-based cohort in the USA, women were found to be at increased risk of LGA in the second pregnancy if their pre-pregnancy BMI category changed towards overweight or obese from first to second pregnancy regardless of their BMI category in first pregnancy except in underweight women who increased to normal weight.[21] This study is different to ours in that it only examined risk in second pregnancy

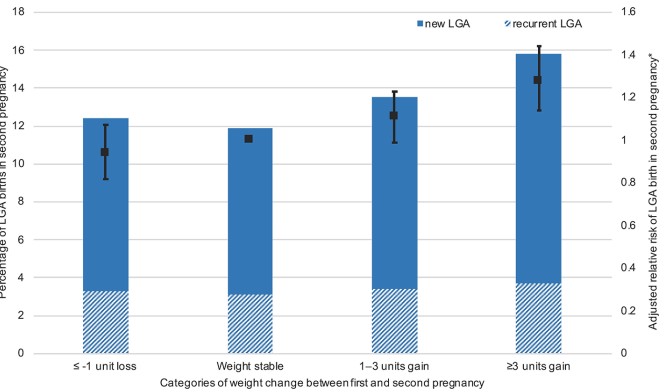

**Figure 3** The percentage and risk of large-for-gestational age (LGA) births in second pregnancy stratified by maternal interpregnancy weight change categories. *Relative risk adjusted for maternal age, ethnicity, highest educational qualification, whether undergone infertility treatment, smoking status, employment status, baseline BMI, gestational diabetes in current pregnancy and interpregnancy interval. BMI, body mass index.

**Table 2** Associations between risk of recurrent large-for-gestational age (LGA) birth in the second pregnancy and change in maternal body mass index (BMI) between pregnancies as measured at the first antenatal visit of each pregnancy stratified by BMI category in the first pregnancy

| Maternal BMI change (categorised) | | Full sample | | | Normal weight at first pregnancy | | | Overweight at first pregnancy | | | Obese at first pregnancy | | |
|---|---|---|---|---|---|---|---|---|---|---|---|---|---|
| | | Total n, No of cases | Relative risk (RR)* | 95% CI | Total n, No of cases | RR* | 95% CI | Total n, No of cases | RR* | 95% CI | Total n, No of cases | RR* | 95% CI |
| Total unadjusted n, No of cases | | 1109, 530 | | | 521, 234 | | | 338, 170 | | | 236, 122 | | |
| Lost ≤ −1kg/m² from previous pregnancy | Unadjusted | 188, 83 | 0.89 | 0.74 to 1.08 | 45, 17 | 0.80 | 0.54 to 1.20 | 74, 30 | **0.68** | **0.50 to 0.94** | 69, 36 | 1.16 | 0.79 to 1.69 |
| | Adjusted† | 178, 78 | 0.88 | 0.72 to 1.07 | 44, 16 | 0.79 | 0.54 to 1.17 | 68, 27 | **0.69** | **0.48 to 0.97** | 66, 35 | 1.21 | 0.79 to 1.83 |
| Weight stable (>−1 to <1kg/m²) | Unadjusted | 365, 181 | Ref | | 212, 100 | Ref | | 98, 58 | Ref | | 51, 23 | Ref | |
| | Adjusted† | 353, 176 | Ref | | 204, 96 | Ref | | 97, 57 | Ref | | 49, 23 | Ref | |
| Gained 1–3kg/m² from previous pregnancy | Unadjusted | 313, 150 | 0.97 | 0.83 to 1.13 | 162, 74 | 0.97 | 0.78 to 1.21 | 90, 43 | 0.81 | 0.62 to 1.06 | 55, 31 | 1.25 | 0.85 to 1.83 |
| | Adjusted† | 301, 142 | 0.98 | 0.84 to 1.15 | 156, 70 | 1.02 | 0.83 to 1.27 | 86, 40 | 0.81 | 0.61 to 1.08 | 53, 30 | 1.28 | 0.86 to 1.91 |
| Gained ≥3kg/m² from previous pregnancy | Unadjusted | 243, 116 | 0.96 | 0.81 to 1.14 | 102, 43 | 0.89 | 0.68 to 1.17 | 76, 39 | 0.87 | 0.66 to 1.14 | 61, 32 | 1.16 | 0.79 to 1.71 |
| | Adjusted† | 234, 111 | 1.00 | 0.83 to 1.20 | 96, 39 | 0.91 | 0.68 to 1.21 | 73, 38 | 0.91 | 0.67 to 1.25 | 61, 32 | 1.28 | 0.84 to 1.94 |

*Generalised linear model with log link and robust variance estimator used to derive RR.
†Adjusted for maternal age, ethnicity, highest educational qualification, whether undergone infertility treatment, smoking status, employment status, baseline BMI, gestational diabetes in current pregnancy and interpregnancy interval.
Bold fonts indicate statistical significance at 0.05 level.

**Table 3** Associations between the risk of new large-for-gestational age (LGA) birth in the second pregnancy following a non-LGA birth in the first pregnancy and change in maternal body mass index (BMI) between pregnancies measured at the first antenatal visit of each pregnancy stratified by BMI category in the first pregnancy

| Maternal BMI change (categorised) | | Full sample | | | Underweight at first pregnancy | | | Normal weight at first pregnancy | | | Overweight at first pregnancy | | | Obese at first pregnancy | | |
|---|---|---|---|---|---|---|---|---|---|---|---|---|---|---|---|---|
| | | Total n, No of cases | Relative risk, (RR)* | 95% CI | Total n, No of cases | RR* | 95% CI | Total n, No of cases | RR* | 95% CI | Total n, No of cases | RR* | 95% CI | Total n, No of cases | RR* | 95% CI |
| Total unadjusted n, No of cases | | 14788, 1573 | | | 606, 24 | | | 8888, 812 | | | 3458, 454 | | | 1836, 283 | | |
| Lost ≤ −1 kg/m² from previous pregnancy | Unadjusted | 2351, 232 | 1.05 | 0.91 to 1.22 | | – | – | 1163, 85 | 0.88 | 0.68 to 1.14 | 690, 79 | 0.95 | 0.73 to 1.24 | 477, 68 | 0.90 | 0.67 to 1.23 |
| | Adjusted† | 2258, 222 | 0.94 | 0.80 to 1.10 | – | – | – | 1108, 81 | 0.87 | 0.68 to 1.12 | 663, 76 | 0.96 | 0.72 to 1.29 | 466, 65 | 0.95 | 0.67 to 1.34 |
| Weight stable (>−1 to <1 kg/m²) | Unadjusted | 5411, 508 | Ref | | 244, 7 | Ref | | 3680, 305 | Ref | | 1024, 123 | Ref | | 463, 73 | Ref | |
| | Adjusted† | 5191, 489 | Ref | | 234, 7 | Ref | | 3519, 292 | Ref | | 985, 118 | Ref | | 453, 72 | Ref | |
| Gained 1–3 kg/m² from previous pregnancy | Unadjusted | 4122, 450 | **1.16** | **1.03 to 1.31** | 230, 8 | 1.21 | 0.45 to 3.29 | 2606, 259 | **1.20** | **1.02 to 1.40** | 888, 127 | 1.19 | 0.94 to 1.50 | 398, 56 | 0.89 | 0.65 to 1.23 |
| | Adjusted† | 3944, 427 | 1.13 | 0.99 to 1.28 | 222, 7 | 1.04 | 0.36 to 3.04 | 2497, 251 | **1.26** | **1.06 to 1.50** | 839, 115 | 1.16 | 0.89 to 1.50 | 386, 54 | 0.86 | 0.61 to 1.22 |
| Gained ≥3 kg/m² from previous pregnancy | Unadjusted | 2904, 383 | **1.40** | **1.24 to 1.59** | 111, 9 | **2.83** | **1.08 to 7.40** | 1439, 163 | **1.37** | **1.14 to 1.64** | 856, 125 | 1.22 | 0.96 to 1.53 | 498, 86 | 1.10 | 0.82 to 1.46 |
| | Adjusted† | 2822, 364 | **1.34** | **1.17 to 1.54** | 104, 6 | 2.08 | 0.67 to 6.51 | 1389, 151 | **1.34** | **1.09 to 1.65** | 839, 123 | **1.35** | **1.05 to 1.75** | 490, 84 | 1.21 | 0.89 to 1.65 |

*Generalised linear model with log link and robust variance estimator used to derive RR.
†Adjusted for maternal age, ethnicity, highest educational qualification, whether undergone infertility treatment, smoking status, employment status, baseline BMI, gestational diabetes in current pregnancy and interpregnancy interval.
Bold fonts indicate statistical significance at 0.05 level.

without adjustment for LGA outcome in first pregnancy. It also considered weight change as change in BMI category only, while we studied change in maternal BMI regardless of whether BMI category has changed or not in the second pregnancy.

In obese women in the USA, interpregnancy weight gain of $\geq 2\,kg/m^2$ was associated with increased risk of LGA and a weight loss of $\geq 2\,kg/m^2$ was associated with decreased risk compared with the reference group of weight maintained (between $> -2$ and $<2\,kg/m^2$).[22] We found no association between weight change and risk of second pregnancy LGA in women who were obese at the start of their first pregnancy. This may be because obese women are already at increased risk of LGA births, and the average interpregnancy BMI change in this subgroup was not large enough to detect a further increase in risk. Greater efforts are needed for primary prevention of obesity in women of childbearing age and obese women need more effective weight loss strategies in interpartum period to assess impact on LGA and other outcomes.

Risk of recurrent LGA was analysed in one previous study in Scotland which found that interpregnancy weight gain ($\geq 2\,kg/m^2$) was associated with increased risk of recurrent LGA. In that study, weight loss ($\geq 2\,kg/m^2$) was associated with reduced LGA risk. Stratification by first pregnancy BMI showed that women with BMI $<25\,kg/m^2$ were at increased risk of recurrent LGA on gaining $\geq 2\,kg/m^2$, whereas women with BMI $\geq 25\,kg/m^2$ were at reduced risk of recurrent LGA on losing $\geq 2\,kg/m^2$ weight.[26] We showed a similar reduction in risk in overweight women who lost $\geq 1$ BMI unit between pregnancies, but found no association in normal weight women. This difference in findings may be because the $<25\,kg/m^2$ group in the previous Scottish study included underweight women whereas our stratified analysis examined normal weight women separately to underweight women.

We showed an increased risk of new LGA in the second pregnancy (after a non-LGA birth in the first pregnancy) with interpregnancy weight gain compared with remaining weight stable. After stratification by BMI, we found that this association between interpregnancy weight gain and new LGA remained only in normal weight and overweight women. The findings from this study are in line with findings with other studies in Scotland[24] and Sweden[23] which found increased risk of new LGA with modest ($1-3\,kg/m^2$) and large ($\geq 3\,kg/m^2$) weight gain. Both studies also found a decreased risk with interpregnancy weight loss of $>1\,kg/m^2$ which was not found in our study. Both studies stratified BMI as $<$ and $\geq 25\,kg/m^2$, while we further stratified the $\geq 25\,kg/m^2$ category as overweight (BMI $25-29.9\,kg/m^2$) and obese ($\geq 30\,kg/m^2$) and found an increased risk of new LGA in overweight, but not in obese women. We carried out sensitivity analysis merging overweight and obese categories and found increased risk in this category (data not shown) suggesting that the results are comparable to previous studies.

Women included in this analysis had a range of interpregnancy interval of $<1$ to up to 12 years and thus weight change could be due to postpartum weight retention or late postpartum weight gain. There is evidence that women who do not lose pregnancy weight at 1 year postpartum are more likely to retain weight longer term.[32] We examined the risk of maternal interpregnancy weight gain with length of the interpregnancy interval and found that women with an interval of 12–23 months were least likely to start the next pregnancy at a higher weight.[33] We also examined the length of the inter-pregnancy interval as a predictor for LGA risk adjusting for interpregnancy weight change and found no association.[34]

The development origins of health and disease concept suggests that adverse exposures during development could lead to enhanced susceptibility in the fetus thus increasing the risk of non-communicable diseases in later life. Although the focus has previously been on exposures during pregnancy, the importance of the preconception period is now recognised.[35–37] Efforts to systematically identify women in the preconception period to improve health and lifestyle during conception are underway.[37] Promoting health of all women of childbearing age with targeting of women and partners planning a pregnancy has been identified as an effective approach to improving preconception health.[36] It is difficult to identify all women who are planning a pregnancy but as the interconception period is also the preconception period for the next pregnancy, it is important to engage with women during this period to optimise their and their children's health.

Future research that characterises the predictors of postpartum weight change would help design interventions to support postpartum weight loss and prevent weight gain. Key to this is an understanding of the pattern of weight change during this period as well as identifying the optimal setting and delivery of the intervention. Support with healthy eating and physical activity is more commonly received during pregnancy than after birth. Even when lifestyle advice is received postpartum, it was found not to be associated with healthy diet or physical activity behaviours.[38] Most interventions that have been successful in limiting and promoting postpartum weight loss were combined diet and physical activity interventions with self-monitoring.[39] However, the timing of engaging women and length of intervention or engagement are important with one study showing that an intervention from 16 weeks' pregnancy to 6 months' postpartum was more effective than the same intervention from birth to 6 months' postpartum intervention.[40]

As pregnancy and early postpartum is a period of major change for women and their families, interventions need to be carefully designed to be attractive, flexible, affordable and feasible for women at this stage with competing priorities and time demands. Focus during the postpartum period in the UK healthcare system is mostly on child health and development. The feasibility and effectiveness of better utilising contact time with health professionals during the 2 years after birth to engage and support maternal health needs to be explored. There may also be a role for peer support groups for mothers. There

is additionally a need to recognise that weight management issues are greater in more disadvantaged mothers so there is also the issue of identifying the most effective weight management strategies for such mothers to reduce social inequity in subsequent birth and maternal outcomes. Weight gain does not occur in isolation and usually combined with other risk factors particularly in socioeconomically disadvantaged groups and hence a holistic approach taking into account priority setting for these families should be considered.

### Strengths and limitations

This is a relatively large population-based cohort including women from all socioeconomic and ethnic backgrounds delivering at a large maternity centre in Southampton, UK, thus representative of the regional population. According to the UK Department of Communities and Local Government English indices of deprivation report, Southampton is more deprived than average with the situation having worsened between 2010 and 2015[41]. However, about half of the women included in this analysis reside in the rest of Hampshire (the region where Southampton is situated), which is less deprived. Our sample was 87% of White ethnicity, which is comparable to the 2011 England and Wales population census of 86% White.[42] The analysis was adjusted for several key confounders that were reasonably complete (96% complete for ethnicity and employment status). Both the maternal weight (used to calculate exposure) and birth weight in this study were objectively measured by healthcare professionals as part of routine antenatal and delivery care.

An important limitation was the lack of information on gestational weight gain during pregnancy, breastfeeding duration/exclusivity and paternal characteristics/behaviour, which are potential confounders in the association between maternal interpregnancy weight gain and LGA birth.[43] We adjusted for if first feed was breast milk as a proxy for breastfeeding initiation in sensitivity analysis and the results remained unchanged (data not shown). Women who had their first booking appointment later into the pregnancy (>24 weeks) were excluded from the analysis in order to ensure comparability of weight measurements between pregnancies. We also adjusted for gestational age at booking, as this was the point when maternal BMI was measured, in sensitivity analysis and the estimates remained similar. Some of the confounding factors which were accounted for in the analysis were self-reported; however, the information was collected prospectively, therefore any measurement error in likely to be non-differential. Another limitation is that these findings are based on observational data so inferences about causation cannot be drawn and the risk of residual confounding influencing the results needs to be considered.

In conclusion, maternal weight gain of 1 or more kg/$m^2$ between first and second pregnancy had a prevalence of 48%, and it was associated with risk of LGA in the second pregnancy in this English cohort. Risk of new LGA was higher in normal weight and overweight women who gained weight after a non-LGA birth in their first pregnancy compared with those who remained weight stable. Overweight women were at a lower risk of a recurrent LGA birth in their second pregnancy if they lost weight between pregnancies. Greater efforts are needed for primary prevention of overweight and obesity in women of childbearing age. Supporting efforts to lose weight in overweight and obese women between pregnancies, and stop weight gain in all women planning to have further children (except those who are underweight) are important preventive measures of subsequent adverse maternal and offspring health outcomes.

**Acknowledgements** We would like to thank David Cable (Electronic Patient Records Implementation and Service Manager) and Florina Borca (Senior Information Analyst R&D) at University Hospital Southampton for support in accessing the data used in this study. We also thank clinical colleagues at the Princess Anne Hospital, Southampton for collecting the data.

**Contributors** Study design (NZ, PJR, NSM, NAA), data analysis (NZ, SW), acquisition and interpretation of the data (NZ, NAA), drafting of the manuscript (NZ), revising for content (NZ, SW, PJR, NSM, NAA) and approval of final version before submission (NZ, SW, PJR, NSM, NAA).

**Funding** This work is supported by a University of Southampton Primary Care and Population Sciences PhD studentship (to NZ), and an Academy of Medical Sciences and Wellcome Trust Grant [AMS_HOP001\1060] (to NAA). NAA is also in receipt of research support from the National Institute for Health Research through the NIHR Southampton Biomedical Research Centre.

**Competing interests** None declared.

**Patient consent for publication** Not required.

**Ethics approval** Approval was granted by the University of Southampton Faculty of Medicine Ethics Committee (ID 25508) and the Health Research Authority (HRA) approval (IRAS 242031).

**Provenance and peer review** Not commissioned; externally peer reviewed.

**Data sharing statement** Anonymised data are only available upon request from the authors conditional on approval of the appropriate institutional ethics and research governance processes.

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
