## [Reviewer comments · BMJ Open]

ARTICLE DETAILS

TITLE (PROVISIONAL)	Is maternal weight gain between pregnancies associated with risk of large-for-gestational age birth? Analysis of a UK population-based cohort
AUTHORS	Ziauddeen, Nida; Wilding, Sam; Roderick, Paul; Macklon, Nicholas; Alwan, Nisreen

VERSION 1 - REVIEW

REVIEWER	A. Bogaerts KU Leuven, Belgium
REVIEW RETURNED	03-Oct-2018

GENERAL COMMENTS	Maternal obesity is increasing worldwide, and represents a considerable health burden for the mother and her child. This study is a valuable study considering the impact of maternal weight gain between pregnancies on the health of future generations. In general the whole paper reads a bit difficult due to complicated sentence constructions. We would suggest to re-write the results section in the abstract to be more attractive for your readership. In your strengths and limitations : What about the possible impact of increasing evolution of pre-pregnancy BMI in relation to 14 years of data collections? Can you comment on this in discussion part? You also state that exposures and outcomes were measured objectively, what about height as this was self-reported, please adapt. Introduction: Paragraph 1 and 2 in the introduction overlap partly, please re-write this as a more integrated part. You don't mention the impact of gestational weight in the second alinea re LGA, please insert state-of-the art here. Third para, line 24, what do you mean here with short and on line 25 with long intervals? Can you be more specific? This also sounds contradictory with the next sentence, has the length of the interval an influence on whether or not the women returns to her pre-pregnancy weight? Third para, line 27: ... six kilograms,... this reads not fluent and is confusing, what does this mean? Can you reformulate this for your readership please? The mentioned six and ten kg in this sentence is also difficult to compare with the BMI units. What is the influence of losing six kg or gaining 10kg on someone's BMI? Line 42 in introduction : which confounders? Methods: Line 21: what is meant by unfeasible?
---

Line 31: so not all exposures are then objectively reported, please adapt this in strengths and limitations summary, as well as on page 9.

Covariables, line 3-5: what is your rationale for dividing between Asian and Chinese? Please comment on this.

Covariables: Gestational diabetes is not mentioned as a covariate, but the models are adjusted for gestational diabetes, please add to the methods.

Statistical analysis, line 23, 24: subsample with women who gained more than or equal to 1 unit BMI and then you write this as a continuous variable : this is unclear and seems contradictory to me, clarify this.

Results:

Line 17: description here is confusing from table 1 as it seems that both groups, those with more as well as those with less than 1 BMI change were more likely to be smokers, please re-formulate.

Line 20: re maternal age: Is there also a sign diff in mean maternal age between those who lost 1 or more versus those who gained 1 or more BMI?

Line 28: "Birthweight (grams) was significantly higher in babies born to women who gained weight between pregnancies (3517g, SD 45) compared to those born to women who lost weight and remained weight stable where the mean birthweight was comparable (3463g, SD 563, 3467g, SD 523 respectively) (p<0.001)." was does 54 gram difference means from a clinical point of view, can you comment on this in discussion please?

Figure 1 is a bit confusing to me and difficult to read, I would suggest not to use stacked columns. + please show which columns are significantly different from each other

Page 6, line 35: why are unadjusted odds ratio's given?

Page 6, line 56: these numbers cannot be found in table 3

Page 7, line 3: from table 3 I would interpret that compared to the reference group normal weight women are at reduced risk if they lost -1BMI unit (unadjusted model, 0,79 p=0,04). Please add this to the results section.

Table 1: Indicate which results are significantly different from each other (e.g. age: lost -1 BMI = 28,7, gained 1 BMI= 28,4 □ is this a significant difference?) Is there a post-hoc analysis done for the ANOVA analysis?

Discussion:

In general, it seems that the authors didn't look into detail in many recent relevant existing studies about impact of interpregnancy weight gain and gestational weight gain (GWG) as the ref list is based on rather old studies. By a quick search on pubmed there are similar studies reporting on this research question, see :

- <https://www.ncbi.nlm.nih.gov/pubmed/29866719> (SR and MA!)
- <https://www.ncbi.nlm.nih.gov/pubmed/28588113>
- <https://www.ncbi.nlm.nih.gov/pubmed/27567535>

- <https://www.ncbi.nlm.nih.gov/pubmed/27165646>
- <https://www.ncbi.nlm.nih.gov/pubmed/27100521>
- <https://www.ncbi.nlm.nih.gov/pubmed/30165842> (GWG and outcomes)

Please use these interesting recent articles if relevant to discuss the results of this analysis more into detail.

Paragraph 1 does not read fluent and is a bit confusing where you repeat results re overweight women and gaining more than or equal to 1 BMI unit. Please re-write this and focus on most clinical important results. Be sufficient critical about differences between BMI groups.

Second para reads difficult, be clear about 'this cohort' do you mean your own analysis or those from other studies?

Can you comment on possible influence of gestational weight gain? Although not available in this analysis?

Page 8 line 8-15: this is rather an enumeration of study results which reads difficult without critical analysis and discussing possible reasons behind, please re-formulate this and try to be more critical to your own results. What would be a possible explanation that there are no associations found?

Page 8, line 34-38: what does this mean clinically?

Page 8/9, line 55-... 4: when and why are contacts planned, what is aim, focus, ..

The discussion is rather superficial. It lacked a thorough and critical analysis of other potential confounders during this typical period in a mother's (family) lifetime. We miss discussion on possible influence of gestational weight gain and postpartum weight retention. What about breastfeeding behavior, possible bariatric surgery in the mother, influence of paternal behavior,... can authors discuss on this too please? Authors state that a lifestyle intervention is of importance, can they discuss also the importance of the preconception period? I miss this link, as most epidemiological studies report on the importance of a healthy pre-pregnancy BMI in relation to outcomes, it even seems more important (DOHaD hypothesis) to start interventions as early and possible and before conception (epigenetics !). The postpartum period can be the preconception period of a next pregnancy, so talking about inter-pregnancy seems clinically an interesting time point. See also the recently started inter-act study protocol, a multicentre study from Belgium (<https://www.ncbi.nlm.nih.gov/pubmed/28549455>).

Strengths and limitations:

You write about a various socioeconomic and ethnic background, .. but your sample consisted mainly of white women (88%) which is rather highly educated,.. can you explain here?

I would have been more commented on the normal weight group as this seems a modifiable group where you can create more impact due to their increased asso with interpreg weight gain and LGA risk, as they are 36% , a not insignificant number of women!

	Minor remarks:  • Abstract: first sentence sounds weird, better high pre-pregnancy BMI and/or excessive gestational weight gain,.. • Line 35 in introduction : in this study ... re-write and be clear which study you refer to from the beginning here • Do we miss refs 23 and 24 in introduction? (page 3, line 49) • Line 37: unclear if 'this' is meant to be the authors own data. • Sometimes numbers are with or without decimal numbers, be consistent
--	---

REVIEWER	Rolv Skjaerven Department of Global Public Health and Primary Care, University of Bergen, Norway
REVIEW RETURNED	11-Oct-2018

GENERAL COMMENTS	The topic of weight gain in relation to pregnancies is important. There are papers with good data and clear results published in high impact journals from other populations/data sources, for instance a Swedish paper from 2006 (reference 23). It is valuable to evaluate whether the same results are found in other populations. The authors have a material that is 10% in size of the Swedish paper. Still they do multivariate analyses adjusting for a myriad of factors in multiple combinations. With much smaller material and accounting for more than a dozen factors the power is reduced (and the confidence intervals are wider). To me, the net result is an unclear study, a set of tables that is almost impossible to read. I really don't understand why not copy the design of reference 23? With less data, why stratify results into 4 categories instead of 2? Also, why not use exposure categories that has worked in the previous papers. 'Gained ≥ 1 BMI unit'? This correspond to approximately 2.7-3kg for a woman with 65kg and height 168cm. Much of the action must be with changes corresponding to 1-3, and 3+ units. In Table 2 the authors study variation in birthweight, and in Model 1 they adjust for gestational age. You should not do that, basically since a 28 week pregnancy can never have a birthweight of 3500grams, still a 28 week pregnancy can have a deviation from the mean of that duration corresponding to 3SD (in my own data we are talking 1000grams with SD=250grams). This analysis has no value, and it become even worse in Model 2 where they in addition adjust for birthweight and gestational age of the first child. If you want to evaluate birthweight by gestational age you can use z-scores of birthweight by gestational age, and probably to stratify by term and preterm births would be good. At what gestational age is it possible to observe a fetal growth effect of a high maternal BMI? I will strongly recommend to provide graphs for the presentation of results, and have a limited set of tables to support the pictures. What is the purpose of providing analyses of LGA, adjusting for previous LGA, and thereafter have analyses on recurrent LGA? Also, technically, you can not use logistic regression estimating ORs for the recurrence since these events typically have occurrences between 30% and 50% (i.e. certainly not rare events). Use relative risk models, for instance as available in STATA.
---

	With Table 3, 4 and 5 the authors present to us more than 100 p-values. Most of these are very much non-significant. In fact, it is not necessary to provide p-values when you give confidence intervals. I applaud the topic to evaluate the impact of BMI increase in successive pregnancies in this population, but the analyses and the presentation are in strong need for reevaluation. Again, much is gained following the analytical strategies, and clear presentation of results, of Villamor and Cnattingius.
--	--

VERSION 1 – AUTHOR RESPONSE

Reviewer: 1

Reviewer Name: A. Bogaerts

Institution and Country: KU Leuven, Belgium Please state any competing interests or state 'None declared': None declared

Please leave your comments for the authors below Thank you for revising the article about "Is maternal weight gain between pregnancies associated with risk of large-for-gestational age birth? Analysis of a UK population-based cohort."

Maternal obesity is increasing worldwide, and represents a considerable health burden for the mother and her child. This study is a valuable study considering the impact of maternal weight gain between pregnancies on the health of future generations.

In general the whole paper reads a bit difficult due to complicated sentence constructions.

Thank you, we have tried to replace long sentences with shorter ones for the purpose of clarity.

We would suggest to re-write the results section in the abstract to be more attractive for your readership.

Thank you, this has now been re-written.

In your strengths and limitations : What about the possible impact of increasing evolution of pre-pregnancy BMI in relation to 14 years of data collections? Can you comment on this in discussion part?

Thank you, we have discussed the differences in prevalence in maternal BMI in comparison with older studies (lines 348-359).

The percentage of obesity over the years for first pregnancy has shown a small increase over time in our cohort but overall the percentage remains comparable over the years (obesity was 11% in 2003 and 14% in 2014 for first pregnancy whereas mean BMI was 24.2 ± 4.2 in 2003 and 24.8 ± 4.9 in 2014).

You also state that exposures and outcomes were measured objectively, what about height as this was self-reported, please adapt.

Thank you - this has been amended to objective measurement of weight (lines 74 and 462).

Introduction:

Paragraph 1 and 2 in the introduction overlap partly, please re-write this as a more integrated part. You don't mention the impact of gestational weight in the second alinea re LGA, please insert state-of-the-art here.

Thank you. Paragraphs 1 and 2 have now been combined into one integrated paragraph with (we hope) clearer sentence structure (lines 83 to 101).

Third para, line 24, what do you mean here with short and on line 25 with long intervals? Can you be more specific?

The length of intervals has been added to clarify (lines 113-115).

This also sounds contradictory with the next sentence, has the length of the interval an influence on whether or not the women returns to her pre-pregnancy weight?

Yes, it does but this paper carried out analysis within and between women and this sentence refers to the analysis that compares women who gained weight to women who did not. We have restructured the paragraph so we hope it now reads more clearly (lines 110-119).

Third para, line 27: ... six kilograms,... this reads not fluent and is confusing, what does this mean? Can you reformulate this for your readership please? The mentioned six and ten kg in this sentence is also difficult to compare with the BMI units. What is the influence of losing six kg or gaining 10kg on someone's BMI?

Thank you, an example has been provided between brackets to clarify what this weight change would mean in BMI units (lines 121-123).

Line 42 in introduction : which confounders?

Thank you, this sentence has now been restructured and merged with the paragraph above.

Methods:

Line 21: what is meant by unfeasible?

What we considered as 'unfeasible' values for weight, height and gestational age have been stated (lines 169-170).

Line 31: so not all exposures are then objectively reported, please adapt this in strengths and limitations summary, as well as on page 9.

Thank you. This has been adapted in the strengths and limitations summary (line 74) and in the discussion (line 462).

Covariables, line 3-5: what is your rationale for dividing between Asian and Chinese? Please comment on this.

Thank you. The categorisation was used as recorded in the hospital system. Given the small proportion of Chinese women in the sample, we have now combined this with the Asian category.

Covariables: Gestational diabetes is not mentioned as a covariate, but the models are adjusted for gestational diabetes, please add to the methods.

Thank you. This has now been added to the methods (line 211 to 215).

Statistical analysis, line 23, 24: subsample with women who gained more than or equal to 1 unit BMI and then you write this as a continuous variable: this is unclear and seems contradictory to me, clarify this.

The subsample was chosen as women who gained ≥ 1 BMI unit. In the analysis itself, the gain in BMI units as a continuous variable was also used to assess the association per unit change in maternal BMI in the women who gained.

We have now removed this table/analysis with continuous BMI change as a predictor from the paper so the presented analysis is only for the categorised version of the exposure. If you/the editor(s) wish, we can include the continuous exposure analysis as a supplementary table.

Results:

Line 17: description here is confusing from table 1 as it seems that both groups, those with more as well as those with less than 1 BMI change were more likely to be smokers, please re-formulate..

Thank you. This has now been clarified. Women who gained ≥ 3 kg/m² are more likely to be smokers compared to the other groups.

Line 20: re maternal age: Is there also a sign diff in mean maternal age between those who lost 1 or more versus those who gained 1 or more BMI?

We have added a sentence about mean age for who had lost weight in the results section (lines 276-277).

Line 28: "Birthweight (grams) was significantly higher in babies born to women who gained weight between pregnancies (3517g, SD 45) compared to those born to women who lost weight and remained weight stable where the mean birthweight was comparable (3463g, SD 563, 3467g, SD 523 respectively) ($p < 0.001$)."

Does 54 gram difference mean from a clinical point of view, can you comment on this in discussion please?

Birthweight is only presented in the descriptive table as we have now removed the regression analysis of birthweight as a continuous outcome from the results and thus we have not commented on this in the discussion section. We are happy to include it as a supplementary table if you/the editor(s) recommend to do so.

Figure 1 is a bit confusing to me and difficult to read, I would suggest not to use stacked columns. + please show which columns are significantly different from each other

Thank you. The figure has been amended and stacked columns are no longer presented. This is now Figure 2. We have added a new Figure 1 (replacing what was previously Table 3) and have presented stacked columns as we also wanted to present adjusted relative risks.

Page 6, line 35: why are unadjusted odds ratio's given?

Thank you. All ORs have now been replaced by RRs based on reviewer two's suggestion.

Page 6, line 56: these numbers cannot be found in table 3

This table and relevant results paragraph have now been removed from the results. We are happy to include it as a supplementary table if you/the editor(s) recommend to do so.

Page 7, line 3: from table 3 I would interpret that compared to the reference group normal weight women are at reduced risk if they lost -1BMI unit (unadjusted model, 0,79 $p = 0,04$). Please add this to the results section.

As above, this table and relevant results paragraph have now been removed from the results so this has not been added.

Table 1: Indicate which results are significantly different from each other (e.g. age: lost -1 BMI = 28,7, gained 1 BMI= 28,4 □ is this a significant difference?) Is there a post-hoc analysis done for the ANOVA analysis?

Thank you, p-values have been included in table 1 to show difference between groups but we have not reported any further post-hoc analysis outside the remit of our research question.

Discussion:

In general, it seems that the authors didn't look into detail in many recent relevant existing studies about impact of interpregnancy weight gain and gestational weight gain (GWG) as the ref list is based on rather old studies. By a quick search on pubmed there are similar studies reporting on this research question, see :

•

<https://emea01.safelinks.protection.outlook.com/?url=https%3A%2F%2Fwww.ncbi.nlm.nih.gov%2Fpubmed%2F29866719&data=01%7C01%7CN.A.Alwan%40soton.ac.uk%7C59eab79f6c1a492614fd08d6463d15f6%7C4a5378f929f44d3ebe89669d03ada9d8%7C1&sdata=kj0LR7KgeuRdW3atzzjslywpbxRbdJJAERdgzmR1U%3D&reserved=0> (SR and MA!)

Thank you. The four studies that assessed LGA as an outcome and were included in this review were included as individual references.

•

<https://emea01.safelinks.protection.outlook.com/?url=https%3A%2F%2Fwww.ncbi.nlm.nih.gov%2Fpubmed%2F28588113&data=01%7C01%7CN.A.Alwan%40soton.ac.uk%7C59eab79f6c1a492614fd08d6463d15f6%7C4a5378f929f44d3ebe89669d03ada9d8%7C1&sdata=b6v%2FZvnUTA%2FwnaZJeQNdw8nRrPvVj%2BtCo4Clx1pL538%3D&reserved=0>

Slightly different outcome of macrosomia (>4 kg).

•

<https://emea01.safelinks.protection.outlook.com/?url=https%3A%2F%2Fwww.ncbi.nlm.nih.gov%2Fpubmed%2F27567535&data=01%7C01%7CN.A.Alwan%40soton.ac.uk%7C59eab79f6c1a492614fd08d6463d15f6%7C4a5378f929f44d3ebe89669d03ada9d8%7C1&sdata=WJ5tFUop354HZ9%2Br%2FITSCxAjhWMugMY3rYdEA39jAmE%3D&reserved=0>

Thank you. This is listed as a reference (25 in the reference list).

•

<https://emea01.safelinks.protection.outlook.com/?url=https%3A%2F%2Fwww.ncbi.nlm.nih.gov%2Fpubmed%2F27165646&data=01%7C01%7CN.A.Alwan%40soton.ac.uk%7C59eab79f6c1a492614fd08d6463d15f6%7C4a5378f929f44d3ebe89669d03ada9d8%7C1&sdata=UChlm5kgZMarZU4SBIS7WjMI%2F87THtUA5Ce7yE3gDDk%3D&reserved=0>

Thank you. LGA is not considered as an outcome in relation to maternal BMI change in this paper

•

<https://emea01.safelinks.protection.outlook.com/?url=https%3A%2F%2Fwww.ncbi.nlm.nih.gov%2Fpubmed%2F27100521&data=01%7C01%7CN.A.Alwan%40soton.ac.uk%7C59eab79f6c1a492614fd08d6463d15f6%7C4a5378f929f44d3ebe89669d03ada9d8%7C1&sdata=SoSFivYeNyUEt676D5DC%2Fg6KIW%2BeGmwMQ%2FjsGM6SxJs%3D&reserved=0>

Thank you. This paper aimed to identify the characteristics associated with recurrent LGA in obese women. A previous publication from the same group using the same cohort assessing the impact of inter-pregnancy weight change on birthweight in obese women was included (reference 22).

•

<https://emea01.safelinks.protection.outlook.com/?url=https%3A%2F%2Fwww.ncbi.nlm.nih.gov%2Fpubmed%2F30165842&data=01%7C01%7CN.A.Alwan%40soton.ac.uk%7C59eab79f6c1a492614fd08d6463d15f6%7C4a5378f929f44d3ebe89669d03ada9d8%7C1&sdata=N7RIGpJw8Pc%2FZ1JqAawubgmLg%2ByQTO0bsl%2BT8NhbQbY%3D&reserved=0> (GWG and outcomes)

Although we recognise the importance of gestational weight gain (GWG) as a predictor, this is a different exposure and was not available in the database, and thus we have not drawn any comparisons to the literature regarding GWG.

Please use these interesting recent articles if relevant to discuss the results of this analysis more into detail.

Many thanks for your suggestions. We have outlined responses above for each suggested reference.

Paragraph 1 does not read fluent and is a bit confusing where you repeat results re overweight women and gaining more than or equal to 1 BMI unit. Please re-write this and focus on most clinical important results. Be sufficient critical about differences between BMI groups.

Thank you. The first paragraph of the discussion section has been re-written based on your recommendation.

Second para reads difficult, be clear about 'this cohort' do you mean your own analysis or those from other studies?

Thank you. This has now been changed to 'our cohort'.

Can you comment on possible influence of gestational weight gain? Although not available in this analysis?

Thank you, we comment on this under limitations of this study (lines 465-468)

Page 8 line 8-15: this is rather an enumeration of study results which reads difficult without critical analysis and discussing possible reasons behind, please re-formulate this and try to be more critical to your own results. What would be a possible explanation that there are no associations found?

Thank you. We have reformulated this section with reflection on the potential reason why our results might be different. We hope it now reads more clearly (lines 381-391)

Page 8, line 34-38: what does this mean clinically?

We have amended text (lines 413-416).

Page 8/9, line 55-... 4: when and why are contacts planned, what is aim, focus, ..

Thank you. We have rephrased this paragraph, and we hope it now reads more clearly.

The discussion is rather superficial. It lacked a thorough and critical analysis of other potential confounders during this typical period in a mother's (family) lifetime. We miss discussion on possible influence of gestational weight gain and postpartum weight retention.

Thank you, we have included discussion on GWG (lines 465-468) and postpartum weight retention (lines 406-410 and 429-440). We have taken your comments into consideration and we hope the discussion now reads better overall.

What about breastfeeding 4, possible bariatric surgery in the mother, influence of paternal behavior,... can authors discuss on this too please?

Thank you, we have included breastfeeding and paternal behaviour discussion points in the strengths and limitations section of the discussion (lines 465-468)

Authors state that a lifestyle intervention is of importance, can they discuss also the importance of the preconception period? I miss this link, as most epidemiological studies report on the importance of a healthy pre-pregnancy BMI in relation to outcomes, it even seems more important (DOHaD hypothesis) to start interventions as early and possible and before conception (epigenetics !). The postpartum period can be the preconception period of a next pregnancy, so talking about inter-pregnancy seems clinically an interesting time point. See also the recently started inter-act study protocol, a multicentre study from Belgium (<https://emea01.safelinks.protection.outlook.com/?url=https%3A%2F%2Fwww.ncbi.nlm.nih.gov%2Fpubmed%2F28549455&data=01%7C01%7CN.A.Alwan%40soton.ac.uk%7C59eab79f6c1a492614fd08d6463d15f6%7C4a5378f929f44d3ebe89669d03ada9d8%7C1&sdata=stDyYfXsIRHLQ%2B4eH88mzN8nNJ0ely0gcc2gBNx6NR4%3D&reserved=0>).

Thank you, we have now included a paragraph in the discussion on DOHaD and preconception (lines 417-428).

Strengths and limitations:

You write about a various socioeconomic and ethnic background, .. but your sample consisted mainly of white women (88%) which is rather highly educated,.. can you explain here?

Thank you. Although the sample has a large proportion of white women, this is representative of this population in Southampton, and the UK as a whole. Considering all women aged 18-39 years, Southampton has a slightly higher proportion of White women (83.4%) compared to the England average (80.5%). Southampton has a slightly higher proportion of Chinese and lower proportion of Black/African women but the ethnicity split remains comparable to the national average¹.

In terms of education, Southampton has a lower proportion of secondary (GCSE) and university education than the national average with a higher proportion of college educated women².

However, as our sample consists of all women who received antenatal care and delivered in the area, this sample is representative of the local population of women becoming pregnant.

We have commented on these issues under 'strengths and limitations' in the discussion section (lines 455-460)

I would have been more commented on the normal weight group as this seems a modifiable group where you can create more impact due to their increased asso with interpreg weight gain and LGA risk, as they are 36% , a not insignificant number of women!

Thank you. We recognise this point therefore we have conducted and reported our stratified analysis using the normal weight subgroup.

Minor remarks:

- Abstract: first sentence sounds weird, better high pre-pregnancy BMI and/or excessive gestational weight gain,...

This has been amended to maternal overweight and obesity. We have not used excessive gestational weight gain as this is not considered in the study.

- Line 35 in introduction : in this study ,.. re-write and be clear which study you refer to from the beginning here

The wording 'in this study' has been removed from this line (line 129).

- Do we miss refs 23 and 24 in introduction? (page 3, line 49)

These have now been added.

- Line 37: unclear if 'this' is meant to be the authors own data.

We think this refers to Page 7 and have amended this to clarify (line 350).

- Sometimes numbers are with or without decimal numbers, be consistent

Thank you. We have checked to confirm that mean/median/standard deviation and percentages are all presented to one decimal place and relative risks to two decimal places. In the discussion when comparing to the literature, we have sometimes used percentages to zero decimal places as this has is how it has been reported in the previous publications.

Reviewer: 2

Reviewer Name: Rolv Skjaerven

Institution and Country: Department of Global Public Health and Primary Care, University of Bergen, Norway Please state any competing interests or state 'None declared': None declared

Please leave your comments for the authors below The topic of weight gain in relation to pregnancies is important. There are papers with good data and clear results published in high impact journals from other populations/data sources, for instance a Swedish paper from 2006 (reference 23). It is valuable to evaluate whether the same results are found in other populations.

The authors have a material that is 10% in size of the Swedish paper. Still they do multivariate analyses adjusting for a myriad of factors in multiple combinations. With much smaller material and accounting for more than a dozen factors the power is reduced (and the confidence intervals are wider)

Thank you. We acknowledge that our sample size is smaller than the Swedish study³. However, it is of comparable size to other published studies in this area^{4,5}. The confidence intervals of the estimates generated from our analyses are not much wider for the multivariable compared to the univariable models.

To me, the net result is an unclear study, a set of tables that is almost impossible to read. I really don't understand why not copy the design of reference 23? With less data, why stratify results into 4 categories instead of 2? Also, why not use exposure categories that has worked in the previous papers. 'Gained ≥ 1 BMI unit'? This correspond to approximately 2.7-3kg for a woman with 65kg and height 168cm. Much of the action must be with changes corresponding to 1-3, and 3+ units.

We have amended tables and added figures to clarify and present the results in a clearer manner. We have presented unadjusted and fully adjusted models only without step-wise adjustment, as the changes are small between the models. We have also split the exposure into categories of 1-3 and ≥ 3 kg/m² BMI change between pregnancies. However, we have retained the baseline BMI categories as

we think it is important to present the difference between overweight and obese women particularly as antenatal care guidelines in the UK recommend encouraging weight loss in women with a pre-pregnancy BMI of ≥ 30 . We show increased risk of LGA in overweight women who gained ≥ 3 kg/m² which would place them either in the top end of the overweight or in the obese category.

In Table 2 the authors study variation in birthweight, and in Model 1 they adjust for gestational age. You should not do that, basically since a 28 week pregnancy can never have a birthweight of 3500grams, still a 28 week pregnancy can have a deviation from the mean of that duration corresponding to 3SD (in my own data we are talking 1000grams with SD=250grams). This analysis has no value, and it become even worse in Model 2 where they in addition adjust for birthweight and gestational age of the first child.

If you want to evaluate birthweight by gestational age you can use z-scores of birthweight by gestational age, and probably to stratify by term and preterm births would be good. At what gestational age is it possible to observe a fetal growth effect of a high maternal BMI?

We have now removed this table with continuous outcome of birthweight and focused the paper on the outcome of LGA only for the sake of clarity of presentation and structure of the paper, and following your recommendation below to reduce the number of tables.

I will strongly recommend to provide graphs for the presentation of results, and have a limited set of tables to support the pictures.

Thank you, we have done that. We now have 3 tables and 2 figures.

What is the purpose of providing analyses of LGA, adjusting for previous LGA, and thereafter have analyses on recurrent LGA? Also, technically, you can not use logistic regression estimating ORs for the recurrence since these events typically have occurrences between 30% and 50% (i.e. certainly not rare events). Use relative risk models, for instance as available in STATA.

Thank you, as above we have added figures and removed two tables to help clarify the results. We have also changed the format of the tables to present a clearer picture. We have updated the analysis to present relative risk for all outcomes as suggested.

With Table 3, 4 and 5 the authors present to us more than 100 p-values. Most of these are very much non-significant. In fact, it is not necessary to provide p-values when you give confidence intervals.

Thank you, we have removed the p-values for tables 2 and 3 and only present RR and confidence intervals. We have retained the P values in table 1 based on comments from reviewer one.

I applaud the topic to evaluate the impact of BMI increase in successive pregnancies in this population, but the analyses and the presentation are in strong need for reevaluation. Again, much is gained following the analytical strategies, and clear presentation of results, of Villamor and Cnattingius.

Thank you for your helpful comments. We have revised the paper and the analysis taking these into account and we hope it now reads better. We believe it is important to publish this analysis as the findings would be generalizable to the UK population given it is based on population-level data

VERSION 2 – REVIEW

REVIEWER	Rolv Skjærven Department of Global Public Health and Primary Care, University of Bergen, Norway
REVIEW RETURNED	30-Jan-2019

GENERAL COMMENTS	Provided the authors add my suggestion to improve Tables 2 and 3, hopefully a little more creative design and content for the figures, Review: The authors have done a good job with this revision. The paper is much easier to read, especially the tables, and they have included graphs, which I recommended. Still I am critical. I hope I said this last time: the tables need to provide number of woman as well as cases for the different categories and strata that they present. By simply listing relative risk (which is much more relevant than the previous odds ratios!) the reader have less information than he normally would like to see. For each of the RR values presented in Table 2 and 3, the authors need to add numbers of women and cases. One value of that is not to recalculate the observed RRs that already are listed, but for the reader to be able to compare rates between the 4 BMI categories (or 3 categories, table 2). For instance build RR with one reference (“weight stable” and “normal weight at first”) and 16 linked RRs (one reference and 15 estimates). I applaud graphs, but the level of creativity in the design and content has to be improved. The value with graphs is to clarify the research idea visually, but also to present the data beyond that of the tables. As for now, the two graphs hold little or no value beyond the tables. The graphs should give the reader additional important information on the topic. May be the strategy mentioned above with one reference category and 15 estimated RRs could be an idea to build from?
--

VERSION 2 – AUTHOR RESPONSE

Review:

The authors have done a good job with this revision. The paper is much easier to read, especially the tables, and they have included graphs, which I recommended.

Still I am critical. I hope I said this last time: the tables need to provide number of woman as well as cases for the different categories and strata that they present.

By simply listing relative risk (which is much more relevant than the previous odds ratios!) the reader have less information than he normally would like to see.

For each of the RR values presented in Table 2 and 3, the authors need to add numbers of women and cases. One value of that is not to recalculate the observed RRs that already are listed, but for the reader to be able to compare rates between the 4 BMI categories (or 3 categories, table 2).

Thank you, we have now added total number of women and number of cases for each categorisation in Tables 2 and 3.

For instance build RR with one reference (“weight stable” and “normal weight at first”) and 16 linked RRs (one reference and 15 estimates).

I applaud graphs, but the level of creativity in the design and content has to be improved. The value with graphs is to clarify the research idea visually, but also to present the data beyond that of the tables. As for now, the two graphs hold little or no value beyond the tables.

The graphs should give the reader additional important information on the topic. May be the strategy mentioned above with one reference category and 15 estimated RRs could be an idea to build from?

Thank you, we have retained Figure 1 (now Figure 3) as we feel this gives the overall picture. We have deleted the previous Figure 2 and added two new figures which presents beyond the information in the tables. We have also included forest plots for Table 2 and Table 3 as supplementary material.